# New Canonical and Grand Canonical Density of States Techniques for Finite Density Lattice QCD

**Christof Gattringer** \*,† , **Michael Mandl** † and **Pascal Törek** †

Institute of Physics, University of Graz, 8010 Graz, Austria; mi.mandl@gmx.at (M.M.);
pascal.toerek@uni-graz.at (P.T.)
\* Correspondence: christof.gattringer@uni-graz.at
† Member of NAWI Graz.

**Abstract:** We discuss two new density of states approaches for finite density lattice QCD (Quantum Chromo Dynamics). The paper extends a recent presentation of the new techniques based on Wilson fermions, while here, we now discuss and test the case of finite density QCD with staggered fermions. The first of our two approaches is based on the canonical formulation where observables at a fixed net quark number $N$ are obtained as Fourier moments of the vacuum expectation values at imaginary chemical potential $\theta$. We treat the latter as densities that can be computed with the recently developed functional fit approach. The second method is based on a direct grand canonical evaluation after rewriting the QCD partition sum in terms of a suitable pseudo-fermion representation. In this form, the imaginary part of the pseudo-fermion action can be identified and the corresponding density may again be computed with the functional fit approach. We develop the details of the two approaches and discuss some exploratory first tests for the case of free fermions where reference results for assessing the new techniques may be obtained from Fourier transformation.

**Keywords:** lattice QCD; finite density; density of states techniques

## 1. Introduction

One of the major open challenges for numerical lattice field theory is the treatment of QCD (Quantum Chromo Dynamics) at finite density. The central problem is the fact that at finite density, the fermion determinant is complex and cannot be used as a probability in Monte Carlo simulations. Density of states (DoS) techniques have been among the possible strategies for overcoming the complex action problem since the pioneering days of lattice QCD [1–6]. The key challenge for DoS techniques is accuracy, since for computing observables, the density needs to be integrated over with a highly oscillating factor. A simple sampling of the density with histogram techniques will allow one to access only very low densities.

An important step for the further development of DoS techniques was presented in [7] where, based on ideas from statistical mechanics [8], a suitable parameterization of the density combined with restricted vacuum expectation values was used to improve the accuracy for the determination of the density of states considerably. In a subsequent series of papers, this so-called LLR method was developed further and assessed for several test cases [9–16]. A related DoS technique, the so-called functional fit approach (FFA), was proposed in [17] and successfully tested in [18–21].

However, all these DoS techniques were formulated for bosonic systems, and no approach to finite density lattice QCD with modern DoS techniques had been presented. Finally, in [22], two possible formulations of DoS techniques for lattice field theories with fermions were suggested. One of the two formulations is the canonical DoS approach (CanDoS) where the density is computed as a function of the imaginary chemical potential $\mu \equiv i\theta/\beta$, where $\beta$ is the inverse temperature. The canonical partition

sum and observables are then obtained as Fourier moments of the density, and the FFA can be used to obtain sufficient accuracy also for the highly oscillating integrals for the higher Fourier modes at large net particle numbers.

The second DoS approach presented in [22] is a direct grand canonical DoS formulation (GCDoS) based on rewriting the grand canonical partition sum of lattice QCD with a suitable pseudo-fermion representation and identifying the imaginary part of the action in this representation. Subsequently, FFA can be applied to evaluate the density as a function of the imaginary part, and again, suitable integrals over the density give rise to vacuum expectation values of observables.

In [22], the two new DoS approaches were presented for the formulation of lattice QCD with Wilson fermions, and the first tests were presented for free Wilson fermions at finite density. In this paper, we now discuss the CanDos formulation and the direct GCDoS approach for the formulation of lattice QCD with staggered fermions. For the CanDos approach, we also present some exploratory tests in the free case, which allows one to assess the accuracy of the method with exact results and to explore the parameters of the new techniques.

## 2. The Canonical Density of States Approach

In this section, we present the basic formulation of the canonical DoS approach (CanDos) for finite density lattice QCD. We stress, however, that the CanDoS approach can easily be implemented for other fermionic theories, e.g., theories with four Fermi interactions generated with auxiliary Hubbard–Stratonovich fields.

### 2.1. Canonical Ensemble and Density of States

We study lattice QCD in $d$ dimensions with two degenerate flavors of quarks. The canonical partition sum at a fixed net quark number $N$ is given by:

$$Z_N = \int_{-\pi}^{\pi} \frac{d\theta}{2\pi} \int \mathcal{D}[U] \, e^{-S_G[U]} \, \det D[U, \mu]^2 \Big|_{\mu = i\frac{\theta}{\beta}} e^{-i\theta N}, \tag{1}$$

where $S_G[U]$ is the Wilson gauge action (we dropped the constant additive term),

$$S_G[U] = -\frac{\beta_G}{3} \sum_{x, \nu < \rho} \text{Re Tr} \, U_\nu(x) \, U_\rho(x + \hat{\nu}) \, U_\nu(x + \hat{\rho})^\dagger \, U_\rho(x)^\dagger. \tag{2}$$

$\beta_G$ is the inverse gauge coupling, and the path integral measure $\mathcal{D}[U]$ in (1) is the product of Haar measures for the link variables $U_\nu(x) \in \text{SU}(3)$. We already integrated out the fermions and obtained the fermion determinants for the two flavors. $D[U, \mu]$ is the Dirac operator at finite chemical potential $\mu$. In this study of the canonical DoS approach, we use the staggered Dirac operator, but stress that it is straightforward to implement the formalism also for different discretizations of the Dirac operator, e.g., for Wilson fermions (compare [22]). The staggered Dirac operator $D[U, \mu]$ is given by:

$$D[U, \mu]_{x,y} = m \, \delta_{x,y} \, \mathbb{1}_3 + \frac{1}{2} \sum_{\nu=1}^{d} \eta_\nu(x) \left[ e^{\mu \, \delta_{\nu,d}} U_\nu(x) \, \delta_{x+\hat{\nu},y} - e^{-\mu \, \delta_{\nu,d}} U_\nu(x - \hat{\nu})^\dagger \, \delta_{x-\hat{\nu},y} \right], \tag{3}$$

where $\eta_\nu(x) = (-1)^{x_1 + \cdots + x_{\nu-1}}$ are the staggered sign factors and $\mathbb{1}_3$ is the unit matrix in color space. We work on a $d$-dimensional lattice of size $N_S^{d-1} \times N_T$, where the temporal ($\nu = d$) extent $N_T$ gives the inverse temperature in lattices units, i.e., $\beta = N_T$. All boundary conditions are periodic, except for the anti-periodic temporal ($\nu = d$) boundary conditions for the fermions. $m$ denotes the bare quark mass and $\mu$ the chemical potential.

In order to project the partition function $Z_N$ to fixed net quark number $N$, in (1), the chemical potential $\mu$ is set to $\mu = i\theta/\beta = i\theta/N_T$ and subsequently integrated over the angle $\theta$ with a Fourier factor $e^{-i\theta N}$. This Fourier transformation with respect to the imaginary chemical potential sets the

net quark number to $N$ and thus generates $Z_N$. The corresponding free energy density is defined as $f_N = -\ln Z_N / V$, where $V = N_S^{d-1} N_T$ denotes the $d$-dimensional volume.

Bulk observables and their moments can be obtained as derivatives of $f_N$ with respect to couplings of the theory. A simple example, which we also will consider in our numerical tests below, is the chiral condensate $\langle \overline{\psi}(x)\psi(x) \rangle_N = \partial f_N / \partial m$,

$$\langle \overline{\psi}(x)\psi(x) \rangle_N = -\frac{2}{V}\frac{1}{Z_N} \int_{-\pi}^{\pi} \frac{d\theta}{2\pi} \int \mathcal{D}[U]\, e^{-S_G[U]}\, \det D[U,\mu]^2 \operatorname{Tr} D^{-1}[U,\mu] \Big|_{\mu=i\frac{\theta}{\beta}} e^{-i\theta N}. \quad (4)$$

The mass derivative leads to the insertion of $\operatorname{Tr} D^{-1}[U,\mu]$ in the path integral. Similarly, general vacuum expectation values of some observable $\mathcal{O}$ at fixed net quark number $N$ have the form:

$$\langle \mathcal{O} \rangle_N = \frac{1}{Z_N} \int_{-\pi}^{\pi} \frac{d\theta}{2\pi} \int \mathcal{D}[U]\, e^{-S_G[U]}\, \det D[U,\mu]^2\, \mathcal{O}[U,\mu] \Big|_{\mu=i\frac{\theta}{\beta}} e^{-i\theta N}. \quad (5)$$

The partition sum (1) and the expressions for the vacuum expectation values (5) can be written with suitable densities $\rho^{(J)}(\theta)$, which we define as:

$$\rho^{(J)}(\theta) = \int \mathcal{D}[U]\, e^{-S_G[U]}\, \det D[U,\mu]^2\, J[U,\mu] \Big|_{\mu=i\frac{\theta}{\beta}}, \quad (6)$$

where $J[U,\mu]$ is set to $J[U,\mu] = \mathbb{1}$ for the partition sum and to $J[U,\mu] = \mathcal{O}[U,\mu]$ for the vacuum expectation values of observables. With the densities $\rho^{(J)}(\theta)$, we may express $\langle \mathcal{O} \rangle_N$ and $Z_N$ as:

$$\langle \mathcal{O} \rangle_N = \frac{1}{Z_N} \int_{-\pi}^{\pi} \frac{d\theta}{2\pi} \rho^{(\mathcal{O})}(\theta)\, e^{-i\theta N}, \qquad Z_N = \int_{-\pi}^{\pi} \frac{d\theta}{2\pi} \rho^{(\mathbb{1})}(\theta)\, e^{-i\theta N}. \quad (7)$$

Note that charge conjugation symmetry can be used to show that $\rho^{(\mathbb{1})}(\theta)$ is an even function such that $\rho^{(\mathbb{1})}(\theta)$ needs to be determined only in the range $\theta \in [0,\pi]$, which cuts the numerical cost in half (see, e.g., [22]). A general observable $\mathcal{O}[U,\mu]$ can be decomposed into even and odd parts under charge conjugation such that also here, the corresponding densities $\rho^{(J)}(\theta)$ need to be evaluated only for $\theta \in [0,\pi]$.

Having defined the densities $\rho^{(J)}(\theta)$ and expressed observables in the canonical ensemble as integrals over the densities, we now have to address the problem of finding a suitable representation of the density and how to determine the parameters used in the chosen representation.

## 2.2. Parametrization of the Density

We need to determine the densities $\rho^{(J)}(\theta)$ for different operator insertions $J$ as discussed in the previous section. For notational convenience, in this section, where we now discuss the parameterization of the densities, we denote all densities as $\rho(\theta)$, but stress that we need to determine the parameters of the different $\rho(\theta)$ independently for every choice of $J$.

The densities $\rho(\theta)$ are general functions of $\theta$ in the interval $[0,\pi]$, which for a numerical determination, we need to describe with only a finite number of parameters. To obtain a suitable parameterization, we divide the interval $[0,\pi]$ into $M$ subintervals as,

$$[0,\pi] = \bigcup_{n=0}^{M-1} I_n, \quad \text{with} \quad I_n = [\theta_n, \theta_{n+1}], \quad (8)$$

where $\theta_0 = 0$ and $\theta_M = \pi$. Introducing $\Delta_n = \theta_{n+1} - \theta_n$ for the length of the intervals $I_n$, we find $\theta_n = \sum_{j=0}^{n-1} \Delta_j$ for $n = 0, 1, \dots M$. For the densities $\rho(\theta)$, we now make the ansatz:

$$\rho(\theta) = e^{-L(\theta)}, \tag{9}$$

where the $L(\theta)$ are continuous functions that are piecewise linear on the intervals $I_n$. We use the normalization $L(0) = 0$, which in turn implies $\rho(0) = 1$. Introducing a constant $a_n$ and a slope $k_n$ for the linear function in every interval $I_n$, we may write $L(\theta)$ in the form:

$$L(\theta) = a_n + k_n [\theta - \theta_n], \quad \text{for } \theta \in I_n = [\theta_n, \theta_{n+1}]. \tag{10}$$

Since the functions $L(\theta)$ are normalized to $L(0) = 0$ and are required to be continuous, we can uniquely determine the constants $a_n$ as functions of the slopes $k_n$ and write $L(\theta)$ in the following closed form:

$$L(\theta) = d_n + \theta\, k_n, \quad \theta \in I_n, \quad d_n = \sum_{j=0}^{n-1} [k_j - k_n] \Delta_j \text{ for } n = 0, \dots M, \tag{11}$$

and express the densities $\rho(\theta)$ as:

$$\rho(\theta) = A_n\, e^{-\theta\, k_n}, \quad \theta \in I_n, \quad A_n = e^{-d_n}. \tag{12}$$

Obviously, the parameterized density $\rho(\theta)$ depends only on the $k_n$, i.e., the set of slopes of the linear pieces in the intervals $I_n$. We point out that our parametrization allows one to work with intervals $I_n$ of different sizes $\Delta_n$ such that in regions where the density $\rho(\theta)$ varies quickly, one may choose small $\Delta_n$, while in regions of slow variation, one may save computer time by working with larger $\Delta_n$.

### 2.3. Evaluation of the Parameters of the Density

To compute the slopes $k_n$ that determine the densities, we introduce so-called restricted expectation values $\langle\, \theta\, \rangle_n(\lambda)$ that are defined as:

$$\langle\, \theta\, \rangle_n(\lambda) \equiv \frac{1}{Z_n(\lambda)} \int_{\theta_n}^{\theta_{n+1}} d\theta \int \mathcal{D}[U]\, e^{-S_G[U]}\, \theta\, e^{\theta\lambda}\, \det D[U, \mu]^2\, J[U, \mu] \Bigg|_{\mu = i\frac{\theta}{\beta}}, \tag{13}$$

where again either $J[U, \mu] = \mathbb{1}$ or $J[U, \mu] = \mathcal{O}[U, \mu]$ is chosen, depending on whether the slopes of the density for the partition sum $Z_N$ or the vacuum expectation $\langle \mathcal{O} \rangle_N$ are being computed. The corresponding restricted partition sums $Z_n(\lambda)$ we use in (13) are given by:

$$Z_n(\lambda) \equiv \int_{\theta_n}^{\theta_{n+1}} d\theta \int \mathcal{D}[U]\, e^{-S_G[U]}\, e^{\theta\lambda}\, \det D[U, \mu]^2\, J[U, \mu] \Bigg|_{\mu = i\frac{\theta}{\beta}}. \tag{14}$$

In the restricted expectation values $\langle\, \theta\, \rangle_n(\lambda)$ and the partition sum $Z_n(\lambda)$, the phase angle $\theta$ is integrated only over the interval $I_n$. We have also introduced a free real parameter $\lambda$, which couples to $\theta$ and enters in exponential form. Varying this parameter allows one to explore the $\theta$-dependence of the density in the whole interval $I_n$ fully. Since for imaginary chemical potential $\mu = i\theta/\beta$, the fermion determinant is real and after squaring also positive, the expectation values $\langle\, \theta\, \rangle_n(\lambda)$ can be evaluated without complex action problem in a Monte Carlo simulation as long as the insertions $J$ are real and positive (for general insertions, $J$ needs to be decomposed into pieces that obey positivity). This is

a technical issue that may be solved also in other ways, e.g., for a bounded observable, the addition of a positive constant is a simple option.

The important observation now is that for the parameterization (12) we have chosen for the densities, $\langle\theta\rangle_n(\lambda)$ and $Z_n(\lambda)$ can be computed also in closed form. Writing the partition sum with the density and then inserting the form (12), one obtains:

$$Z_n(\lambda) = \int_{\theta_n}^{\theta_{n+1}} d\theta\,\rho(\theta)\,e^{\theta\lambda} = e^{-d_n}\int_{\theta_n}^{\theta_{n+1}} d\theta\,e^{-\theta k_n}e^{\theta\lambda} = e^{-d_n}\,\frac{e^{\theta_n[\lambda-k_n]}}{\lambda-k_n}\left(e^{\Delta_n[\lambda-k_n]}-1\right). \tag{15}$$

From a comparison of (13) and (14), one finds that the restricted vacuum expectation value $\langle\theta\rangle_n(\lambda)$ can be computed as the derivative $\langle\theta\rangle_n(\lambda) = d\ln Z_n(\lambda)/d\lambda$, such that also $\langle\theta\rangle_n(\lambda)$ can be found in closed form:

$$\langle\theta\rangle_n(\lambda) \equiv \frac{d\ln Z_n(\lambda)}{d\lambda} = \theta_n + \frac{\Delta_n}{1-e^{-\Delta_n[\lambda-k_n]}} - \frac{1}{\lambda-k_n}. \tag{16}$$

Using a multiplicative and an additive normalization, we bring $\langle\theta\rangle_n(\lambda)$ into a standard form $V_n(\lambda)$ where the result is expressed in terms of a simple function $h(s)$,

$$V_n(\lambda) \equiv \frac{\langle\theta\rangle_n(\lambda) - \theta_n}{\Delta_n} - \frac{1}{2} = h\big(\Delta_n[\lambda - k_n]\big) \quad\text{with}\quad h(s) \equiv \frac{1}{1-e^{-s}} - \frac{1}{s} - \frac{1}{2}. \tag{17}$$

The function $h(s)$ obeys $h(0) = 0$, $h'(0) = 1/12$, and $\lim_{s\to\pm\infty} h(s) = \pm 1/2$.

The determination of the slope $k_n$ for the interval $I_n$ now consists of the following steps: For several values of $\lambda$, one computes the corresponding restricted vacuum expectation values $\langle\theta\rangle_n(\lambda)$ defined in (14) and brings them into the normalized form $V_n(\lambda)$ defined in Equation (17). Fitting the corresponding data with $h\big(\Delta_n[\lambda - k_n]\big)$ allows one to determine the $k_n$ from a simple stable one-parameter fit. From the sets of the slopes $k_n$, we can determine the densities $\rho(\theta)$ using (11) and (12) and finally compute the observables via the integrals (7).

## 3. An Exploratory Test of the Canonical DoS Approach in the Free Case

As a first assessment of the new canonical density of states approach, we tested the new method for the case of free fermions at finite density in two dimensions. This served to verify the method and the program and allowed for exploring the parameters of the method, such as the number of intervals $I_n$ and suitable choices for the values of $\lambda$. In addition, for the free case, all steps of the CanDoS approach could be cross-checked with exact results obtained from Fourier transformation.

*3.1. Setting and Reference Results from Fourier Transformation*

For this first test, we used the chiral condensate at fixed particle number $\langle\overline{\psi}(x)\psi(x)\rangle_N = \partial f_N/\partial m$ as our main observable. For the free case, the corresponding expression (4) reduces to:

$$\langle\overline{\psi}(x)\psi(x)\rangle_N = -\frac{2}{V}\frac{1}{Z_N}\int_{-\pi}^{\pi}\frac{d\theta}{2\pi}\,\det D[\mu]^2\,\operatorname{Tr} D^{-1}[\mu]\,\bigg|_{\mu=i\frac{\theta}{\beta}}e^{-i\theta N}, \tag{18}$$

where all links in the Dirac operator (3) were set to $U_\nu(x) = \mathbb{1}$. For implementing the CanDoS approach for the condensate, we need the two densities,

$$\rho^{(\mathbb{1})}(\theta) = \det D[\mu]^2\,\bigg|_{\mu=i\frac{\theta}{\beta}} \quad\text{and}\quad \rho^{(\operatorname{Tr} D^{-1})}(\theta) = \det D[\mu]^2\,\operatorname{Tr} D^{-1}[\mu]\,\bigg|_{\mu=i\frac{\theta}{\beta}}. \tag{19}$$

For determining the slopes $k_n$ of these two densities, we thus have to compute the restricted expectation values (13) for $J = \mathbb{1}$ and $J = \operatorname{Tr} D^{-1}$. Normalizing the corresponding Monte Carlo data

according to (17) and fitting them with $h\big(\Delta_n[\lambda - k_n]\big)$ gives rise to the slopes $k_n$. From the respective sets of slopes, we find the densities $\rho^{(1)}(\theta)$ and $\rho^{(\mathrm{Tr}\, D^{-1})}(\theta)$ using (11) and (12), and finally, the vacuum expectation value $\langle\, \overline{\psi}(x)\psi(x)\,\rangle_N$ is obtained as:

$$\langle\, \overline{\psi}(x)\psi(x)\,\rangle_N \;=\; -\frac{2}{V}\frac{1}{Z_N}\int\limits_{-\pi}^{\pi}\frac{d\theta}{2\pi}\,\rho^{(\mathrm{Tr}\, D^{-1})}(\theta)\,e^{-i\theta N}, \qquad Z_N \;=\; \int\limits_{-\pi}^{\pi}\frac{d\theta}{2\pi}\,\rho^{(1)}(\theta)\,e^{-i\theta N}. \qquad (20)$$

In the free case, the reference results can be obtained with the help of Fourier transformation. Furthermore, for the case of two flavors in two dimensions, which we are using for our test, we can explore the relation $\det D[\mu]^2 = \det D_{naive}[\mu]$ between the determinant of the staggered Dirac operator $D[\mu]$ and the determinant of the naive Dirac operator $D_{naive}[\mu]$, which in two dimensions is given by:

$$D_{naive}[\mu]_{x,y} = m\,\delta_{x,y}\,\mathbb{1}_2\times\mathbb{1}_3 \;+\; \frac{1}{2}\sum_{\nu=1}^{2}\sigma_\nu\times\mathbb{1}_3\left[e^{\mu\,\delta_{\nu,2}}\,\delta_{x+\hat\nu,y} - e^{-\mu\,\delta_{\nu,2}}\,\delta_{x-\hat\nu,y}\right], \qquad (21)$$

where $\sigma_1$ and $\sigma_2$ are the first two Pauli matrices acting on the Dirac indices of the two-component spinors used in the naive discretization and $\mathbb{1}_2$ is the corresponding unit matrix. All link variables were set to their trivial values, i.e., they were replaced by the $3\times 3$ unit matrix $\mathbb{1}_3$. The determinant of the naive Dirac operator can be computed by first diagonalizing $D_{naive}[\mu]$ in space-time with the help of Fourier transformation and then taking the product of the corresponding momentum space Dirac operator determinants over all momenta.

The density $\rho^{(1)}(\theta)$ then was simply obtained via numerically evaluating $\det D_{naive}[\mu]$ for $\mu = i\theta/\beta$. For the density $\rho^{(\mathrm{Tr}\, D^{-1})}(\theta)$, one may use Jakobi's formula ($d\,\det M/dx = \det M\,\mathrm{Tr}[M^{-1}\,dM/dx]$) for the derivative of a determinant and the fact that $dD/dm = \mathbb{1}$ to obtain:

$$\rho^{(\mathrm{Tr}\, D^{-1})}(\theta) = \det D[\mu]^2\,\mathrm{Tr}\,D^{-1}[\mu]\bigg|_{\mu=i\frac{\theta}{\beta}} = \frac{1}{2}\frac{d}{dm}\det D[\mu]^2\bigg|_{\mu=i\frac{\theta}{\beta}} = \frac{1}{2}\frac{d}{dm}\det D_{naive}[\mu]\bigg|_{\mu=i\frac{\theta}{\beta}}. \qquad (22)$$

The vacuum expectation value $\langle\, \overline{\psi}(x)\psi(x)\,\rangle_N$ can be obtained from (20) by numerically integrating over $\theta$. For the reference data in the plots below, we implemented this integration with Mathematica.

### 3.2. Numerical Results for CanDos in the Free Case

Having discussed the observables and the corresponding densities for the free case, as well as the evaluation of reference data with the help of Fourier transformation, we now come to a brief exploratory numerical test for the free case in $d = 2$ dimensions. The results in the plots below were computed on $16\times 16$ lattices at a mass parameter of $m = 0.1$. We used 50 intervals $I_n$ of equal size to parameterize the density in the range $[0,\pi]$. For each interval, we computed the restricted expectation values (16) for 20 different values of $\lambda$ using Monte Carlo simulations based on $10^6$ measurements, where in the simulation, the fermion determinant was evaluated exactly with Fourier transformation. The restricted expectation values were then normalized to the form (17) and the slopes $k_n$ determined from the corresponding fits with $h\big(\Delta_n[\lambda - k_n]\big)$. From the slopes, the densities were computed using (11) and (12).

In Figure 1, we show the results for the densities $\rho^{(1)}(\theta)$ (lhs plot) and $\rho^{(\mathrm{Tr}\, D^{-1})}(\theta)$ (rhs). The thin blue curves are the results from the CanDos determination and the thick magenta curves the reference data computed with Fourier transformation as discussed in the previous subsection. Obviously, the CanDos densities matched the reference data very well.

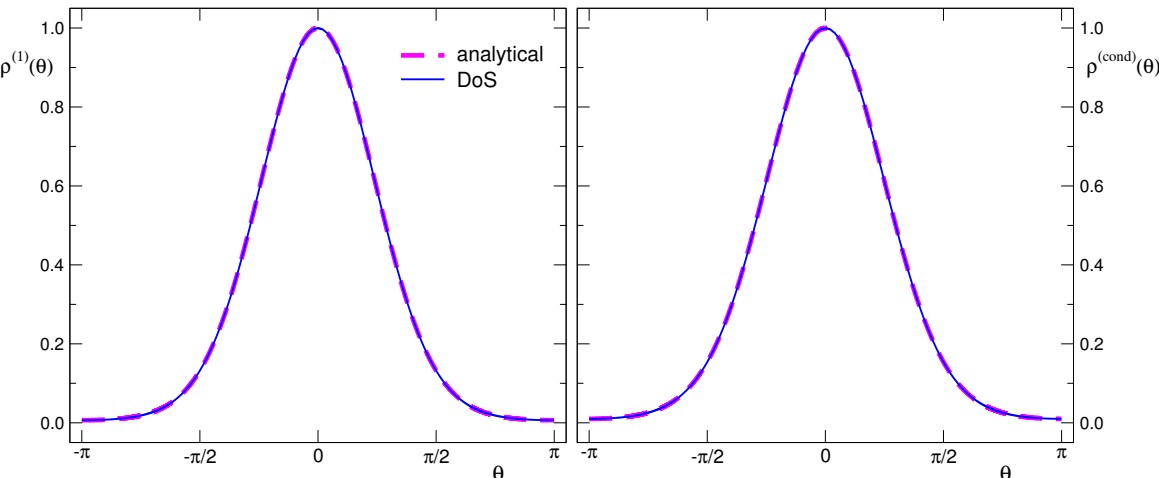

**Figure 1.** The densities $\rho^{(\mathbb{1})}(\theta)$ (lhs) and $\rho^{(\operatorname{Tr} D^{-1})}(\theta)$ (rhs figure; denoted as $\rho^{(\mathrm{cond})}(\theta)$ in the plot). We compare the data from the canonical DoS (CanDoS) determination (thin blue curves) to the analytic results obtained with Fourier transformation (thick dashed magenta curves). The data are for $16 \times 16$ lattices with $m = 0.1$ and densities are normalized to $\rho(0) = 1$.

Having determined the densities, we can compute the canonical partitions sums $Z_N$ and vacuum expectation values at fixed net fermion number using (7). In the lhs plot of Figure 2, we show our results for the canonical partition sums $Z_N$ normalized by $Z_0$ as a function of $N$. The results from the CanDos determination are shown as red dots, the reference data from Fourier transformation as black diamonds. Here as well, we observed essentially perfect agreement for all values of the net fermion number $N$ we considered. A more physical quantity is the corresponding free energy density $f_N = -\ln Z_N / V$ (here normalized to $f_0 = 0$), which in the rhs plot of Figure 2, we show as a function of $N$. Again, we compared the CanDos results (red dots) to the corresponding reference data (black diamonds) and found very good agreement, and only for the largest net particle number $N = 10$ shown in the plot, we observed a slight deviation, indicating that for net quark numbers $N > 10$, the accuracy of the determination of the density would have to be improved, e.g., by using more and finer intervals $I_n$.

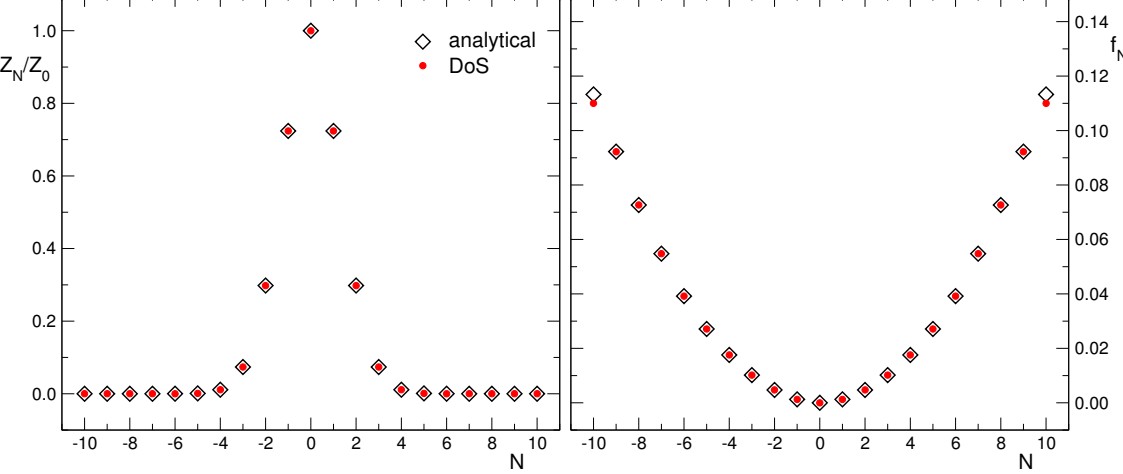

**Figure 2.** The canonical partition sums $Z_N / Z_0$ (lhs) and the corresponding free energy densities $f_N = -\ln(Z_N / Z_0) / V$ (rhs) as a function of the net fermion number $N$. The parameters are $V = 16 \times 16$ with $m = 0.1$, and we compare the results from the CanDoS determination (red dots) to the analytic results obtained with Fourier transformation (black diamonds).

We conclude our exploratory study with discussing the vacuum expectation value of an observable, i.e., a case where the ratio of two integrals over two different densities needs to be computed. The quantity we considered was the chiral condensate, and the two corresponding densities $\rho^{(\mathbb{1})}(\theta)$ and $\rho^{(\operatorname{Tr} D^{-1})}(\theta)$ were the ones already shown in Figure 1. For both of them, we found very good agreement with the reference data, and the crucial question now was if this translated also into the corresponding physical observable matching the reference data well. In Figure 3, we show the CanDos results (red dots) for the condensate $\langle \overline{\psi}(x)\psi(x) \rangle_N$ as a function of the net quark number $N$. Indeed, we found a very satisfactory agreement with the results from Fourier transformation (black diamonds) up to $N = 7$ where the first deviations became visible. Again, for higher values of $N$, a more precise determination of the involved densities will be necessary.

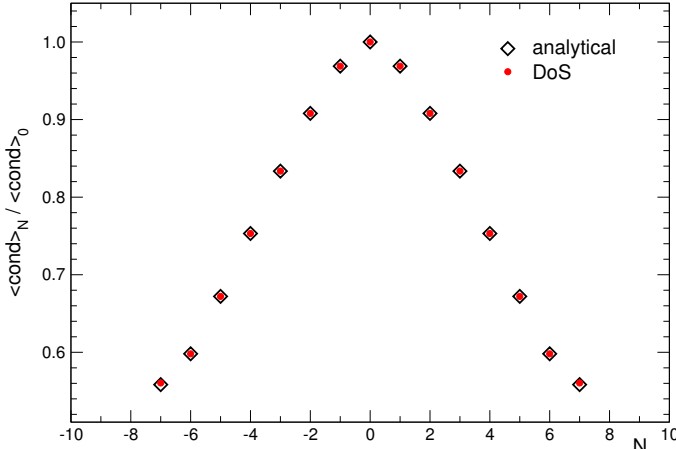

**Figure 3.** The chiral condensate $\langle \overline{\psi}(x)\psi(x) \rangle_N$ (in the plot denoted as $\langle cond \rangle_N$ and normalized by the $N = 0$ value) as a function of the net fermion number $N$. The parameters are $V = 16 \times 16$ with $m = 0.1$, and we compare the results from the CanDoS determination (red dots) to the analytic results obtained with Fourier transformation (black diamonds).

We close the discussion of our numerical test by stressing once more that the results presented here could only be considered a very preliminary assessment of the new CanDos approach. The tests were done in two dimensions, and only the free case was considered (although this already constituted a non-trivial test of the method). Currently, we are extending the assessment of CanDos by implementing a study in 2-dQCD, but also started to explore lattice field theories with four Fermi interactions.

## 4. Direct Grand Canonical DoS Approach

In this section, we now briefly discuss our second DoS approach, which is based on a suitable pseudo-fermion representation of the grand canonical QCD partition sum (GCDoS approach). We will determine the imaginary part of the pseudo-fermion action and set up the FFA to compute the density as a function of the imaginary part.

### 4.1. Pseudo-Fermion Representation and Introduction of Densities

The starting point was the grand canonical partition sum of QCD. We again considered two flavors of staggered fermions such that the grand canonical partition sum at chemical potential $\mu$ is given by:

$$Z_\mu = \int \mathcal{D}[U] \, e^{-S_G[U]} \, \det D[U, \mu]^2, \tag{23}$$

where $S_G[U]$ is again the Wilson gauge action (2), and the staggered Dirac operator $D[U, \mu]$ is specified in (3).

We first identically rewrite the fermion determinant and subsequently express the part with the complex action problem in terms of pseudo-fermions,

$$\det D[U,\mu] = \det(D[U,\mu]D[U,\mu]^\dagger)\frac{1}{\det D[U,\mu]^\dagger} = C \det(D[U,\mu]D[U,\mu]^\dagger) \int \mathcal{D}[\phi]e^{-\phi^\dagger D[U,\mu]^\dagger \phi}, \quad (24)$$

where $C$ is an irrelevant numerical constant and $\phi(x)$ are complex-valued pseudo-fermion fields that have three color components. The measure $\int \mathcal{D}[\phi]$ simply is a product measure where at every site of the lattice, each component is integrated over the complex plane. The overall factor $\det(D[U,\mu]D[U,\mu]^\dagger)$ is obviously real and positive and can be treated with standard techniques [23,24]. The exponent of the pseudo-fermion integral on the other hand has a non-vanishing imaginary part and thus requires a strategy for dealing with the corresponding complex action problem.

To set up the direct DoS approach in the grand canonical formulation, we divided the exponent of the pseudo-fermion path integral into real and imaginary parts,

$$\phi^\dagger D[U,\mu]^\dagger \phi = S_R[\phi,U,\mu] - iX[\phi,U,\mu], S_R[\phi,U,\mu] = \phi^\dagger A[U,\mu]\phi, X[\phi,U,\mu] = \phi^\dagger B[U,\mu]\phi, \quad (25)$$

where we defined two matrices for the kernels of the real and imaginary parts of the pseudo-fermion action,

$$A[U,\mu] = \frac{D[U,\mu]+D[U,\mu]^\dagger}{2}, B[U,\mu] = \frac{D[U,\mu]-D[U,\mu]^\dagger}{2i}. \quad (26)$$

It is straightforward to evaluate $A[U,\mu]$ and $B[U,\mu]$ explicitly,

$$A[U,\mu]_{x,y} = m\delta_{x,y}\mathbb{1} + \frac{1}{2}\sum_{\nu=1}^{d}\eta_\nu(x)\sinh(\mu\delta_{\nu,d})\left[U_\nu(x)\,\delta_{x+\hat{\nu},y} + U_\nu^\dagger(x-\hat{\nu})\,\delta_{x-\hat{\nu},y}\right],$$

$$B[U,\mu]_{x,y} = -\frac{i}{2}\sum_{\nu=1}^{d}\eta_\nu(x)\cosh(\mu\delta_{\nu,d})\left[U_\nu(x)\,\delta_{x+\hat{\nu},y} - U_\nu^\dagger(x-\hat{\nu})\,\delta_{x-\hat{\nu},y}\right]. \quad (27)$$

The fermion determinant thus assumes the form:

$$\det D[U,\mu] = C\,\det(D[U,\mu]D[U,\mu]^\dagger)\int\mathcal{D}[\phi]\,e^{-S_R[\phi,U]+iX[\phi,U]}. \quad (28)$$

We already remarked that the real and positive overall factor $\det(D[U,\mu]D[U,\mu]^\dagger)$ could be treated with conventional simulation techniques [23,24], which we will not address in detail here (see [22] for a discussion of this term in the Wilson fermion formulation). Together with the Boltzmann factor for the gauge field action, we combined this term into a new effective action Boltzmann factor defined as:

$$e^{-S_{eff}[U,\mu]} = e^{-S_G[U]}\,\det(D[U,\mu]D[U,\mu]^\dagger). \quad (29)$$

The grand-canonical partition sum thus can be written as:

$$Z_\mu = \int\mathcal{D}[U]\int\mathcal{D}[\phi]\,e^{-S_{eff}[U,\mu]}\,e^{-S_R[\phi,U,\mu]}\,e^{iX[\phi,U,\mu]}. \quad (30)$$

The next step is to introduce suitable densities for the imaginary part:

$$\rho^{(J)}(x) = \int\mathcal{D}[U]\int\mathcal{D}[\phi]\,e^{-S_{eff}[U,\mu]}\,e^{-S_R[\phi,U,\mu]}\,J[\phi,U,\mu]\,\delta(x - X[\phi,U,\mu]), \quad (31)$$

where we again allow for the insertion of functionals $J[\phi,U,\mu]$ in order to take into account different observables. As for the CanDos approach, one may use charge conjugation symmetry to show that the

densities are either even or odd functions of $x$, depending on the insertion $J[\phi, U, \mu]$ (see [22]). Thus, it is sufficient to compute the densities only for positive $x$.

With the help of the densities vacuum, the expectation values of observables in the grand canonical picture at chemical potential $\mu$ can be written as:

$$\langle \mathcal{O} \rangle_\mu = \frac{1}{Z_\mu} \int_0^\infty dx \, \rho^{(\mathcal{O})}(x) \, e^{ix} \, , \quad Z_\mu = \int_0^\infty dx \, \rho^{(\mathbb{1})}(x) \, e^{ix}. \tag{32}$$

*4.2. Evaluation of the Densities with FFA*

Having defined the densities and expressed grand canonical vacuum expectation values as suitable integrals over the densities, we now can set up the FFA approach for evaluating the densities.

First, we remark that the densities $\rho^{(J)}(x)$ are expected to be fast decreasing functions of $x$, and in [22], this was indeed verified in test cases. Thus, we may cut off the integration range in (32) to a finite interval $[0, x_{max}]$ and determine the density only for this range. As for the canonical case, we divided the interval $[0, x_{max}]$ into $M$ intervals $I_n = [x_n, x_{n+1}]$, $n = 0, 1, \dots M - 1$, with $x_0 = 0$ and $x_M = x_{max}$. As for the CanDos formulation, the densities were parameterized by the negative exponential of a function $L(x)$ that was continuous and piecewise linear on the intervals $I_n$. Again, we assumed the normalization $L(0) = 0$, and the density thus was entirely determined by the slopes $k_n$.

In the FFA approach, the slope $k_n$ in each interval $I_n$ is determined from suitable restricted vacuum values, which we here define as:

$$\langle X \rangle_n(\lambda) = \frac{1}{Z_n(\lambda)} \int \mathcal{D}[U] \int \mathcal{D}[\phi] e^{-S_{eff}[U,\mu]} e^{-S_R[\phi,U,\mu]} e^{\lambda \, X[\phi,U,\mu]} J[\phi, U, \mu] \, \Theta_n \big( X[\phi, U, \mu] \big), \tag{33}$$

where we have defined the support function $\Theta_n(x)$:

$$\Theta_n(x) = \begin{cases} 1 \text{ for } x \in I_n, \\ \phantom{1} 0 \text{ else.} \end{cases} \tag{34}$$

As in the canonical case, also the generalized expectation values (33) can be expressed in terms of the parameterized density and computed in closed form, along the lines discussed above. After normalizing them to the form (17), the generalized expectation values are again described by the functions $h\big( \Delta_n[\lambda - k_n] \big)$ such that the slopes $k_n$ can be determined from one parameter fits. Subsequently, the densities are constructed from the slopes using (11) and (12), with $\theta$ replaced by $x$. Finally we can compute observables from the densities using (32).

The direct, grand canonical density of states approach discussed in this section for staggered fermions was discussed for Wilson fermions in [22]. There, also first exploratory numerical results were presented, and for free fermions it was shown that the density obtained with the FFA approach matched exact reference data from Fourier transformation very well.

**5. Summary and Outlook**

In this paper, we extended our previous work [22], where we presented two new DoS techniques for finite density lattice QCD with Wilson fermions, to the formulation of QCD with staggered fermions. The first formulation was based on the canonical formulation where the canonical partition sum and vacuum expectation values of observables at fixed net quark number were obtained as Fourier moments with respect to imaginary chemical potential. The functional fit approach (FFA) could then be used to compute the density with sufficient accuracy for reliably determining observables for reasonable net quark numbers. We presented exploratory tests of the canonical DoS approach for the case of free fermions in 2-dand found that observables such as the chiral condensate at finite net quark numbers reliably matched reference data obtained from a direct calculation with Fourier transformation that was possible in the free case.

Our second approach was set up directly in the grand canonical ensemble. The QCD partition sum was rewritten in terms of a suitable pseudo-fermion representation, and the imaginary part of the pseudo-fermion action was identified. Using FFA, the density was then computed as a function of the imaginary part, and grand canonical vacuum expectation values were again obtained as the corresponding oscillating integrals. The tests of the new approaches presented here were done for the staggered fermion formulation, but we would like to point out again that also the Wilson formulation could be used and refer to our paper [22] for the discussion of the corresponding results.

Two comments are in order here: Although the first tests were encouraging, the numerical results presented here clearly constituted only a very preliminary and exploratory assessment of the new techniques. We are currently extending these tests towards QCD in two dimensions as the next test case before approaching the full 4-dtheory. We furthermore stress that the techniques we presented here were not restricted to QCD or other gauge theories with fermions. Furthermore, theories with four Fermi interactions could be accessed after the introduction of suitable Hubbard–Stratonovich fields, and also for this direction of possible further development we have started exploratory calculations.

**Author Contributions:** Conceptualization and analytical work: C.G., M.M., and P.T.; software and numerical work: M.M. and P.T.; validation: C.G., M.M., and P.T.; paper writing: C.G. All authors read and agreed to the published version of the manuscript.

**Funding:** This work is supported by the Austrian Science Fund FWF, Grant I 2886-N27, and partly also by the FWF DK 1203 "Hadrons in Vacuum Nuclei and stars".

**Acknowledgments:** We thank Mario Giuliani, Peter Kratochwill, Kurt Langfeld, and Biagio Lucini for interesting discussions.

**Conflicts of Interest:** The authors declare no conflict of interest.

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
