# Peer review of "New Canonical and Grand Canonical Density of States Techniques for Finite Density Lattice QCD"

_2571-712X, doi:10.3390/particles3010008_

Round 1
Reviewer 1 Report
The paper discusses two new approaches to the problem of finite-density QCD based on density-of-states methods. The feasibility of the approaches, destined to avoid the sign problem in QCD, is successfully tested for the case of free fermions in two dimensions: the comparisons of the numerical simulations with analytical results (that are possible to get for the case of free fermions) gave excellent agreements. The methods are thus promising candidates to resolve the long-standing problem, to be applied to the full QCD in the future.
The manuscript may be accepted in the present form with minor technical corrections:
1. DoS and FFA should be de-abbreviated in the title and abstract
2. Section 5: "Two comments are in oder here", the misprint should be corrected.
Author Response
We thank the referee for his/her very careful reading of the manuscript and the feedback. In the revised version the abbreviations DoS and FFA are now spelled out in the title and abstract and the type is fixed.
With best regards, Christof Gattringer (on behalf of all authors)
Reviewer 2 Report
Dear Editor,
In the manuscript the authors discuss 2 new approaches based on density of states techniques used for finite density lattice QCD. The first one is based on the canonical ensemble whereas the second one is based on the grand canonical ensemble. A first preliminary test for the free case in 2 dimensions gives very good agreement with the direct calculation using the Fourier transformation.
The paper is very well written and organized and the results are interesting. After clarifying some minor points and implementing some suggestions for improvement (see below) I believe that the manuscript is suitable for publication in MDPI-Particles.
(1) The authors only used the staggered Dirac operator to test the canonical approach. It would have been interesting to see how the approach performs in a test with a different Dirac operator.
(2) The authors write that “Since for imaginary chemical potential µ = iθ/β the fermion determinant is real and after squaring also positive, the expectation values […] can be evaluated without complex action problem in a Monte Carlo simulation as long as the insertions J are real and positive (for general insertions J needs to be decomposed into pieces that obey positivity).” Does this positivity requirement of J have any deeper physical consequence? Maybe the authors could comment on that.
(3) In Figure 1 I suggest to use not only different colors for the curves but also different styles (e.g. solid and dashed lines).
(4) In line 78 it should be “save” instead of “safe”.
Author Response
We thank the referee for his / her report and the suggestions for improving the paper. Here our reply to the 4 issues raised by the referee and how we included the recommendations in the revised version:
1) The approach was also tested with Wilson fermions and the corresponding results are discussed in our Ref. [22] that just has appeared in Phys. Ref. D. In order to point this out clearer in our presentation we added a corresponding comment in the second paragraph of the summary. Reference [22] now is also updated to the journal reference.
2) The positivity of the observable is merely a technical issue, that can be treated in different ways. In the text we had mentioned a possible splitting into positive and negative parts, but other approaches are possible, such as adding a constant if an observable is bounded. We added a orresponding comment at the end of the paragraph below equation (14).
3) We now use dashed lines for one of the curves and also adapted the caption of the figure accordingly.
4) The typo noted by the referee was fixed.
We thank the referee once more for his/her comments and hope that the paper is ready for publication now.
With best regards, Christof Gattringer (for all authors)